# Validation of a New Contactless and Continuous Respiratory Rate Monitoring Device Based on Ultra-Wideband Radar Technology

**DOI:** 10.3390/s21124027

**Published:** 2021-06-11

**Authors:** Timo Lauteslager, Michal Maslik, Fares Siddiqui, Saad Marfani, Guy D. Leschziner, Adrian J. Williams

**Affiliations:** Circadia Technologies Ltd., 6 Delmey Close, London CR0 5QD, UK; michal@circadia.health (M.M.); fs@circadia.health (F.S.); saad.marfani@gmail.com (S.M.); guy@circadia.health (G.D.L.); adrian@circadia.health (A.J.W.)

**Keywords:** respiratory rate monitor, continuous vital sign monitoring, contactless monitor, remote patient monitoring, ultra-wideband radar, performance validation, analysis of agreement

## Abstract

Respiratory rate (RR) is typically the first vital sign to change when a patient decompensates. Despite this, RR is often monitored infrequently and inaccurately. The Circadia Contactless Breathing Monitor™ (model C100) is a novel device that uses ultra-wideband radar to monitor RR continuously and un-obtrusively. Performance of the Circadia Monitor was assessed by direct comparison to manually scored reference data. Data were collected across a range of clinical and non-clinical settings, considering a broad range of user characteristics and use cases, in a total of 50 subjects. Bland–Altman analysis showed high agreement with the gold standard reference for all study data, and agreement fell within the predefined acceptance criteria of ±5 breaths per minute (BrPM). The 95% limits of agreement were −3.0 to 1.3 BrPM for a nonprobability sample of subjects while awake, −2.3 to 1.7 BrPM for a clinical sample of subjects while asleep, and −1.2 to 0.7 BrPM for a sample of healthy subjects while asleep. Accuracy rate, using an error margin of ±2 BrPM, was found to be 90% or higher. Results demonstrate that the Circadia Monitor can effectively and efficiently be used for accurate spot measurements and continuous bedside monitoring of RR in low acuity settings, such as the nursing home or hospital ward, or for remote patient monitoring.

## 1. Introduction

The importance of respiratory rate (RR) for the assessment of patient health status has long been established [1]. RR as a vital sign is highly sensitive to changes in a patient’s condition and is often the first vital sign to change when a patient decompensates. RR has been shown to be an early indicator for a wide range of complications [2,3,4,5,6]. High RR has also been shown to be a predictor for admission into the intensive care unit [7,8] and for hospital mortality [9]. Despite its obvious significance, RR is frequently monitored imprecisely [10], inaccurately, or not at all [11,12]. In low acuity settings, such as the general ward or nursing home, RR is obtained by manual spot measurements: breaths are visually counted in a 60 s window, or shorter. Notably, healthcare professionals may estimate RR often as either 18 or 20 breaths per minute (BrPM) [13], and large errors are found on manual clinical assessments of RR [14,15]. Additional to the lack of accuracy, manual spot measurements of RR are usually not performed frequently enough for timely detection of adverse events. In the non-intensive ward setting, more than 90% of prolonged hypoxemic episodes may be missed when relying on intermittent spot check monitoring [16].

Alternative respiratory monitoring strategies tethering the patient (e.g., capnography, pulse oximetry, or electrocardiography-derived RR) may be feasible in intensive care, but are not suitable for long term monitoring in lower acuity settings. Technological advances have provided further options for continuous monitoring. Wrist-worn wearable devices or wireless finger probes (such as a pulse oximeter) can estimate RR from heart rate data, obtained through optical sensors [17]. Wearable patches provide greater accuracy [18], albeit at lower levels of patient comfort. Long term monitoring compliance, particularly in nursing facilities or for remote patient monitoring, remains a challenge. Practical matters, such as limited battery life [19] or devices getting misplaced, hamper data collection. Finally, discomfort associated with wearable devices may still be prohibitive for adoption [20]. For robust, long-term vital sign monitoring, devices must be affordable, easy-to-use and unobtrusive [19], and are ideally contactless.

Radar technology has the potential to provide contactless, continuous, and long-term RR monitoring. Displacement of the chest and abdomen wall, caused by respiration, can be sensed by transmitting and receiving radio frequency (RF) signals, allowing for contactless RR assessment. Early radar-based RR monitoring systems often used continuous-wave (CW) Doppler radar [21,22,23]. CW radar systems rely on a continuously transmitted, single-tone RF signal, of which reflections may be phase-modulated by thoracoabdominal motion. Thoracoabdominal motion Δx and RF signal phase variation Δφ are related in the following way:(1)$$\Delta \varphi =\frac{4\pi \;\Delta x}{\lambda }$$
where λ is the wavelength of the transmitted RF single-tone signal. As described in detail in [24], detection of phase variation is typically done by down-conversion to baseband by mixing the received signal with the transmitted signal (and optionally with an out-of-phase component, as proposed by [25]). CW radar systems require a simple architecture and can be built using low-cost components [26], hence the method’s early popularity. However, the method does not allow for detection of the user’s distance, nor for filtering potential noise sources based on proximity to the system. This drawback may be resolved by stepping through a range of single-tone frequencies, essentially synthesizing a time-domain pulse in the frequency domain, with a down-range resolution proportional to bandwidth (more detail in [27]). Such architecture is referred to as stepped-frequency continuous-wave (SFCW) radar, or frequency-modulated continuous-wave (FMCW) radar when a continuous range of frequency is used (such as a chirp signal). Both the SFCW and FMCW architectures have been shown to be successful for RR monitoring [28,29,30], but these systems are substantially more complex and power consuming than the CW radar architecture. An alternative method is ultra-wideband (UWB) radar. UWB radar relies on transmitting short RF pulses and determining the time-of-flight (ToF) of reflected pulses (Figure 1). The distance $D_{0}$ from the radar to the source of the reflection is calculated from ToF:(2)$$D_{o}=\frac{c\;ToF}{2}$$
where c is the speed of light. The respiration-caused chest and abdomen excursion causes small but periodic variations in distance to the reflector (Δx in Figure 1), which may be tracked from periodic ToF variation. When the transmitted RF pulse is not just a single impulse, but instead comprises a transient oscillation with a carrier wave of known phase φ, ToF variations may be tracked from variations to the carrier wave phase (Δφ in Figure 1), in a similar fashion to CW radar. Various methods have been proposed to obtain carrier phase φ from received UWB radar signals, including arctangent demodulation, the Hilbert–Huang transform, and tracking zero crossings [27]. Advantages of using UWB radar for RR monitoring include the ability to determine both RR and distance of the patient, the option of filtering out noise sources based on their proximity to the radar (time-gating), multiple-person monitoring, UWB’s increased resistance to multipath interference [31], and the ability to sense through objects such as walls or rubble (relevant for rescue applications) [32,33,34]. Although UWB radar has been used for RR monitoring [35,36], the requirement of costly and power-hungry equipment (pulse generator and oscilloscope) prohibited widespread use. More recently, integrated circuit technology has made UWB radar available at a fraction of the cost, and in small, power-efficient systems [37,38,39]. UWB radar chips have been adopted for vital sign monitoring, and successful RR monitoring has been demonstrated [37,40,41,42].

Despite impressive progress in radar sensor technology, and the general acceptance of the importance of continuous RR monitoring, radar is currently not being used for continuous monitoring in clinical settings at scale. Most proposed systems are not commercially available, and have not been appropriately validated. For clinical acceptance, it is required that a new monitoring technology is validated by analysis of agreement against a gold-standard reference, across a representative patient sample, under representative conditions.

The current work validates the performance of a new RR monitor, based on UWB radar technology: The Circadia Contactless Breathing Monitor™ (model C100), by Circadia Technologies Ltd. (London, UK). The device is FDA 510(k)-cleared, and can be used for both spot measurements and continuous monitoring of RR. To date, performance validation data of the device have not been available.

## 2. Materials and Methods

To validate the accuracy of the Circadia Monitor, RR as obtained by the device was compared to manually scored data from conventional respiratory monitors, including end-tidal carbon dioxide (ETCO2) capnography, nasal pressure, and respiratory effort belts (measuring motion of the chest and abdomen). ETCO2 capnography is a direct measure of respiration. Respiratory motion, reflected in the respiratory rate, is a result of contraction of the diaphragm resulting in outward movement of the abdomen, along with complementary contraction of accessory thoracic muscles resulting in outward movement of the thoracic cage. The resulting airflow is also reflected precisely by changes in nasal pressure. The employed reference devices are standard in the measurement of disordered breathing, and count as the gold standard of RR measurement when RR is obtained manually from sensor data by a certified professional.

Three separate studies were performed to validate device performance across a range of clinical and nonclinical settings. A total of 50 subjects were included in data collection, amounting to 809 min of analyzed reference RR data.

### 2.1. Study Population and Procedures

All procedures were conducted in accordance with the Declaration of Helsinki, and informed consent was obtained for each subject. Ethical approval of Study 1, Study 2, and Study 3 was obtained from IntegReview IRB (Austin, Texas; Ref. No. 2019-RESPIRE-1), Aga Khan University Ethics Review Committee (Karachi, Pakistan; Ref. No. 2019-1457-5438), and from the institutional review board of the University of Fribourg Department of Psychology (Fribourg, Switzerland; Ref. Nr. 2018-280), respectively. For each of the three studies, anyone meeting one of the following vulnerable population criteria was excluded from participation: anyone below the age of 18, pregnant women, anyone unable or with limited capacity to provide consent, prisoners, terminally ill patients, and employees of either the study sponsor or contract research organization. Subjects with disorders causing uncontrollable, involuntary movements were excluded from participation to avoid movement artefacts. An overview of the subject populations recruited in Studies 1, 2, and 3, as well as an overview of study procedures, is given in Table 1.

All testing conditions comprised the subject sitting still in a chair or lying down on a mattress or bed while recordings were taken using both the C100 Circadia Monitor and a reference device. For Study 1, various recording conditions were performed to simulate typical clinical use cases. For Studies 2 and 3, the Circadia Monitor was placed either beside the bed or behind the bed headboard at a device-to-subject distance (*D*_0_) of approximately 1.0 m (representing a typical long-term monitoring use case), as shown in Figure 2. For all cases, 60 s epochs of reference data were manually scored by a certified technician to obtain reference RR. Procedures for individual studies are described below.

#### 2.1.1. Study 1

Subjects were recruited through online advertisement. To ensure a wide range of subject characteristics, a gender balance was pursued, as well as a uniform distribution of body mass index (BMI; ranging from 18–40), and a uniform distribution of age (18–75). Subjects were generally healthy, although comorbidities were not recorded.

Various spot measurements and one measurement of extended duration were performed to validate performance across use cases, distances *D*_0_, and subject position. Subjects were asked to minimize movement. For each of the following eight conditions, two 1 min spot measurements were obtained:Seated, at 0.5 m distance *D*_0_;Seated, at 1.0 m distance *D*_0_;Seated, at 1.0 m distance *D*_0_, subject covered by blanket;Seated, at 1.5 m distance *D*_0_;Lying down in supine position, at 0.5 m distance *D*_0_;Lying down in supine position, at 0.5 m distance *D*_0_, subject covered by blanket;Lying down on the side, facing towards the device, at 0.5 m distance *D*_0_;Lying down on the side, facing away from the device, at 0.5 m distance *D*_0_.A 9 min, continuous recording was obtained for the following condition:Lying down in supine position, at 0.5 m distance *D*_0_.Thoracic effort, abdominal effort, nasal pressure, and nasal flow were obtained using the Natus Embletta MPR PG (Pleasanton, CA, USA) polygraphy device.

#### 2.1.2. Study 2

Subjects were recruited from a group of patients who were taking part in a sleep assessment at Aga Khan University Hospital (Karachi, Pakistan), for the diagnosis of obstructive sleep apnea (OSA) or for continuous positive airway pressure device titration. Subjects in this sample were generally overweight or obese, and suffered from various health conditions, including OSA, hypertension, and diabetes mellitus.

Overnight polysomnography (PSG) data, including ETCO2 capnography, were collected using a Nihon Kohden PSG device (Tokyo, Japan). For each subject, a 10 min window of PSG data was selected for scoring, with a minimal number of apnea events and motion artefacts. Due to severe OSA and substantial motion by subjects and/or staff, suitable PSG windows were selected manually, while blinded to Circadia data.

#### 2.1.3. Study 3

Subjects were recruited from a group of volunteers taking part in an overnight PSG sleep and memory study at the University of Fribourg (Fribourg, Switzerland). All subjects were young and healthy, recruited from a student population. Each subject completed either one or two overnight recordings.

Overnight PSG data, including abdominal respiratory effort belt data, were collected using a Brain Products PSG device (Gilching, Germany). For each subject, a 10 min window of data was collected for scoring, at which subjects were assumed to be asleep. A total of 15 nights of data were collected and included.

### 2.2. Manual Scoring of Reference Data

Reference PSG data, subdivided in 60 s epochs, were manually scored for RR by a certified technician. In addition, data from Study 1 were visually inspected by a respiratory physiologist. The technician and respiratory physiologist were asked to exclude epochs containing sensor artefacts, or highly irregular breathing patterns with no obvious RR. In Study 2, due to the sample being composed of patients with OSA, epochs containing apnea events were excluded. Both the technician and respiratory physician were blinded to the Circadia data. An example of a 60 s epoch of PSG sensor data, Circadia continuous RR data, manually scored PSG RR, and Circadia mean RR is given in Figure 3, with PSG data in red and Circadia data in blue.

### 2.3. Statistical Analysis

A data analysis and statistical plan for Study 1 was written and filed with a private entity before Study 1 data were accessed. The same plan was applied to data from Study 2 and 3, which were originally collected for different research objectives.

The Circadia Monitor produces an RR value every 3 s. For comparison purposes, a mean RR value was computed for each 60 s epoch. Agreement between 60 s epoch Circadia RR and reference RR data was assessed through the Bland–Altman method. The 95% limits of agreement (LOA) were computed as the bias (mean difference between each method) ±2 standard deviations of the difference between each method. The 95% CI around the 95% LOA was calculated. Within-subject correlation was accounted for when necessary. The 95% CI around the LOA considering multiple observations per subject was computed according to the method by Zou [43]. All statistical analyses were performed in MedCalc (MedCalc Software Ltd., Ostend, Belgium). A target performance was defined as an agreement of ±5 BrPM. Thus, the LOA and 95% CI must fall within this range. The target performance was based on findings that interobserver agreement of RR, in clinical practice, is ±4 to 5 BrPM [14,15].

The number of excluded epochs were counted and reported per recording condition. If no Circadia RR data were present for a valid epoch on reference recording due to, for example, motion artefacts or device failure, the recording was labeled as ‘failed’. The Circadia success rate was calculated as the fraction of epochs for which RR was obtained using the Circadia device. Error between the resulting Circadia RR data and reference RR data was calculated for each 60 s epoch. Mean absolute error (MAE) was computed. Accuracy rate was determined as the percentage of recordings for which the absolute error did not exceed 2 BrPM.

## 3. Results

After withdrawal of one subject, Study 1 was completed by 26 subjects. Study 2 was completed by 12 subjects. Subjects in this sample were generally obese and suffered from various health conditions, including OSA (*n* = 12), hypertension (*n* = 10), diabetes mellitus (*n* = 5), chronic kidney disease, atrial fibrillation, stroke, amyotrophic lateral sclerosis, and hypothyroidism. Study 3 was completed by 12 subjects. Study 1 aimed to cover broad ranges of age and BMI, as well as both genders, to ensure generalizability across user characteristics. The number of subjects per age category was as follows: *n* = 8 (18–34 years), *n* = 6 (35–49 years), *n* = 6 (50–64 years), and *n* = 6 (65–75 years). The number of subjects per BMI category was: *n* = 7 (18–24), *n* = 11 (24–30), *n* = 4 (30–35), and *n* = 4 (35–40). Finally, the number of subjects per gender was: *n* = 10 (female), and *n* = 16 (male).

From a total of 468 recorded epochs in Study 1 (26 subjects, 18 recordings each), 22 recordings failed due to reference device failure, and three recordings were never completed because the subject was not able to sit still. An additional 45 epochs were excluded through data review by both the respiratory physician and technician due to uninterpretable ventilatory effort signals. For seven epochs, the Circadia Monitor was not able to give a reading. A total of 392 epochs were used for further analyses.

In Study 2, a total of 120 epochs were recorded (12 subjects, 10 min of data per subject). Three epochs were excluded for containing artefactual capnography data. Nine more epochs were labeled as containing apnea events and excluded from analysis. For five more epochs, no data were available due to a delayed start of recording equipment. For eight epochs, the Circadia Monitor was not able to give a reading, in part due to incorrect device placement in one of the subjects. A total of 95 epochs were used for further analysis. 

In Study 3, a total of 150 epochs were recorded (15 nights, 10 min of data per night). Six epochs were excluded for containing artefactual respiratory effort belt data.

### 3.1. Overall Analysis of Agreement

Analyses of agreement between Circadia RR (‘Circadia’) and manually scored RR (‘Manual’) were performed on an aggregate of Study 1 spot measurement data, on all Study 2 data, and on all Study 3 data. Only two epochs from the sustained recording condition (condition I) in Study 1 were included in the aggregate to maintain balance between conditions. The resulting Bland–Altman plots are shown in Figure 4. Table 2 provides an overview of results. The total number of epochs, number of epochs that were excluded by the technician or physician, and percentage of epochs for which Circadia data were successfully obtained are denoted for each study. MAE and accuracy rate (for an error margin of 2 BrPM) are given, as well as Bland–Altman analysis results (bias, upper and lower LOA, plus CIs). For mean metrics (success rate, MAE, and accuracy rate), the mean across subjects is given, as well as the standard deviation. 

The 95% LOA for Study 1, 2, and 3 were −3.0 to 1.3 BrPM, −2.3 to 1.7 BrPM, and −1.2 to 0.7 BrPM, respectively. For all three studies, agreement between Circadia and reference method was found to be high and well within the target agreement of ±5 BrPM. No proportional bias was observed. However, a fixed bias ranging from −0.27 to −0.85 BrPM was found, indicating that Circadia RR tends to be slightly higher than manually scored RR. A Circadia success rate of over 90% was found for all three studies, and data could successfully be obtained for all study subjects. Study 2 success rate would have been 95% if excluding subjects for whom a deviation from study protocol was detected; for two subjects, a distance between subject and device exceeding 2 m was found, which is substantially higher than the protocol dictated, and higher than the device’s recommended use.

### 3.2. Analysis of Agreement per Condition

To validate performance across user position and use cases, a breakdown of results per recording condition (A–I) in Study 1 is given in Table 3. The total number of epochs, number of excluded epochs, Circadia success rate, MAE, accuracy rate, and Bland–Altman analysis results are presented.

For all nine conditions, including measurements at a 1.5 m distance *D*_0_, LOA and the CI (not shown for brevity) were found to be within the target agreement of ±5 BrPM and no proportional bias was observed. Again, a fixed bias ranging from −0.6 to −1.0 BrPM was measured, indicating that Circadia RR, on average, exceeded manually scored RR. For all conditions, Circadia success rate was over 90%, and accuracy rate was 85% or higher.

### 3.3. Respiratory Waveform Correspondence

To demonstrate the correspondence between thoracoabdominal excursion as recorded by the Circadia device, and ventilatory effort as recorded by a PSG reference device, recorded raw waveforms are shown in Figure 5 and Figure 6. All four example epochs were obtained in Study 1, from different subjects. Embletta nasal pressure and abdominal effort are shown in red, and the Circadia respiratory waveform is shown in blue. All figures show high waveform correspondence between Circadia and reference device, in particular with the reference abdominal effort belt. Variations in abdominal excursion amplitude, as well as rapid variations in breathing frequency, are accurately captured.

High waveform correspondence will generally lead to good agreement between Circadia RR and manually counted RR, as exemplified by Figure 5a. However, small discrepancies may occur when high levels of breath-to-breath interval variability are present, as illustrated by Figure 5b. The example epochs in Figure 6 both aim to illustrate how brief pauses in breathing, or even the temporary absence of breathing, may cause agreement bias and the occasional large error that is observed when comparing continuous RR to manually scored RR.

## 4. Discussion

Performance of the Circadia Monitor for measurement of continuous RR was assessed by direct comparison to manually scored reference data. Agreement was found to be high for all study data and for all individual experimental conditions. Bland–Altman analysis pointed out that 95% LOA (and their CI) fell within the predefined acceptance criteria of ±5 BrPM. Importantly, the Circadia agreement results exceeded those reported for manual clinical assessment of RR [14,15]. Accuracy rate, using a narrower error margin of ±2 BrPM, was found to be 90% or higher for all study data. Similar levels of agreement and accuracy were found for short (0.5 m), medium (1.0 m), and high (1.5 m) subject-to-device distances, indicating that performance is not affected by distance, at least up to 1.5 m. When comparing performance between conditions with different subject orientation, it was found that accuracy was higher in conditions where the subject faced the Circadia Monitor. As described in [28], chest excursion due to respiration is larger towards the frontal direction; hence, it is expected that the respiratory signal is easier to discern. Despite a small drop in performance, agreement between Circadia RR and reference RR was well within the acceptance criteria for all subject orientations. Slightly reduced performance was observed in the lying down conditions of Study 1, when compared to seated conditions. However, it should be noted that both Study 2 and 3 were performed with subjects lying asleep underneath a blanket, and particularly high accuracy was observed in Study 3. Visual data inspection revealed that the lying down conditions in Study 1 were more prone to reference data artefacts and irregular breathing.

Several contactless RR monitoring devices have been developed (see [44] for a review), but only a subset of these are suitable for continuous monitoring in a healthcare setting, and most are not validated properly. Performance validation should be done against the gold standard, which is manually scored RR. Unfortunately, automatically scored RR is often used as a reference instead, which introduces error and may favor the device under test, which itself relies on automatic scoring. Although validated against automatically scored RR, validation results of three contactless RR monitors are given for comparison.

In a small sample, a radar-based monitor was compared to automatically scored capnography data and 95% LOA of −5.82 to 4.63 BrPM were found [29]. A second study validating the same device, as well as a contactless acoustics-based RR monitor, was performed. Different statistical methods were used to obtain reported results, but Bland–Altman analysis pointed out that 95% LOA for both devices exceeded the ±5 BrPM limit [45]. Finally, a second radar-based RR monitor was validated by comparison to automatically scored PSG data, and found 95% LOA of −4.0 and 3.0 BrPM, and −4.5 and 1.8 BrPM, depending on the reference PSG device [46]. Agreement results for the Circadia Monitor (95% LOA of −3.0 to 1.3 BrPM, −2.3 to 1.7 BrPM, and −1.2 to 0.7 BrPM, for Study 1, 2 and 3, respectively) clearly exceeded published results for other contactless devices.

Circadia results demonstrate that the system can effectively and efficiently be used for both spot measurements and continuous bedside monitoring of RR in low acuity settings, such as the nursing home, hospital ward, or remote patient monitoring. RR data can be recorded whenever a patient is at rest, providing accurate vital sign data throughout the day.

### 4.1. Methodology and Sources of Error

In the manual scoring process, less than 8% of epochs were rejected by the reviewing technician or physician. Epoch rejection corresponds to clinical practice, where only artefact-free sections of data are used for RR scoring. Reasons for rejection were artefactual sensor data, presence of movement artefacts, and highly erratic breathing making reference data uninterpretable. This in itself demonstrates the limitations of current standard technologies for continuous RR monitoring.

Despite high correspondence between Circadia and reference waveforms, small discrepancies in RR may occur when breathing rate changes rapidly. This is illustrated by Figure 5b, where the last section of the epoch shows substantially lower breathing rates. This sudden change strongly affects manually counted RR, but is reflected in Circadia data that falls just outside of the considered epoch. Precision of RR estimates decreases with reduced window length [10], hence discrepancies are likely to occur when validating a continuous monitoring system using spot measurements of limited duration.

Occasionally, measured error exceeded ±2 BrPM. This was typically a result of substantial irregularities and pauses in breathing. Circadia RR values are produced only when breathing is present, and some signal stationarity is assumed. An abrupt interruption of breathing lasting no longer than 1 breath will typically not be counted. When apneas are present, this will result in an overestimation of RR in a 60 s window, when compared to a manual count. Figure 6a,b both exemplify this. Because apnea events were not expected in Study 1, the technician was not asked to exclude epochs containing pauses in breathing. A few subjects, however, showed regular apneas during wake, leading to error in RR estimates. A potential solution to help avoid RR estimation errors in practice would be to visually inspect respiratory waveforms. Alternatively, the use of continuous RR instead of spot measurements would help average out errors, obviating the need for data inspection.

The small but systematic overestimation of RR by the Circadia device when compared to manually scored data (ranging from 0.27 to 0.85 BrPM) is a result of the assumed periodicity in short time windows. For example, when a subject displays regular breathing of 15 BrPM, but skips a single breath, the Circadia device will output a value of 15. However, a manual count in a 60 s window will result in a 14 BrPM. When the reviewer considers a shorter analysis window, such as 30 or 20 s (common in practice), the resulting estimated RR will drop to a value as low as 12 BrPM. This illustrates how RR is not unambiguously defined when respiration is irregular, and shows again that increasing window length through continuous monitoring is advantageous [10].

### 4.2. Study Samples and Generalizability

Study 1 data demonstrate Circadia performance across user characteristics. The nonprobability sample was designed to cover a broad range of age and BMI, as well as both genders. Sample striation targets were set to ensure an approximate uniform distribution. After epoch exclusion, 23 to 25 subjects remained in the Study 1 sample, depending on the experimental condition. Epochs from three subjects, in particular (of different categories across the age, BMI, and gender striation), ended up being excluded for various experimental conditions. Initial striation targets would still be met if these subjects were to be excluded from the sample.

To demonstrate performance in the clinical population, patients taking an OSA assessment were recruited for Study 2. As a result of patients being elderly and often obese, a range of comorbidities was observed. The current results demonstrate Circadia performance on a generally unhealthy and obese population, with health conditions including OSA, hypertension, and diabetes mellitus. It is not expected that different health conditions will entirely mask or stop chest and abdominal motion required for breathing. It was found that irregular breathing patterns can affect the performance of the device, at least when comparing to manually scored data. However, as argued above, it is challenging to define RR unambiguously when respiration is irregular or even temporarily absent. Respiratory waveforms could always be inspected in case of concern of pathological breathing patterns.

Study 3 data demonstrate performance on subjects while asleep. The requirement of manually scoring reference data prohibited accuracy analysis of whole nights. However, across the 8 h recordings, Circadia success rates were high, and over 80% for almost all recordings.

For all three studies, subjects suffering from uncontrollable, involuntary movement disorders were excluded. Given that no factors other than user movement are expected to affect the monitor’s ability to measure RR, study results are thought to be generalizable to the broader population.

### 4.3. Limitations of the Study

The current studies assess accuracy and success rate of RR monitoring. For continuous and unsupervised monitoring, the ability to reject non-respiration data is equally important. The Circadia Monitor is designed to only produce a value for RR when a valid respiration signal is present. User movement can typically be distinguished from a user lying still and breathing, and is normally rejected automatically. Further investigations must aim to validate this ability to ignore potential movement artefacts. Apart from user movement, sources of noise in a clinical setting could include nursing staff, other patients on a ward, or devices such as fans or oxygen concentrators. For contactless devices, such as the Circadia Monitor, it must also be validated that a device does not produce readings when no user is in its detection range.

The Circadia Monitor does not measure respiration directly, but instead estimates RR from abdomen and chest motion. This, of course, can also be said for the majority of respiratory rate monitors, as well as for the industry standard of assessing RR by looking at a patient’s chest wall motion. However, it must be considered that, in case of airway obstruction, chest and abdomen movement may occur without effective respiration.

## 5. Conclusions

Performance of a novel, contactless, continuous respiratory rate monitor was assessed by direct comparison to manually scored reference data. Performance was assessed across various relevant use cases and subject characteristics, in both a clinical and non-clinical setting. RR, as recorded by the Circadia Contactless Breathing Monitor™ (model C100), matched the reference RR data well within acceptable limits, making this novel device a suitable alternative to current measures of RR.

## Figures and Tables

**Figure 1 sensors-21-04027-f001:**
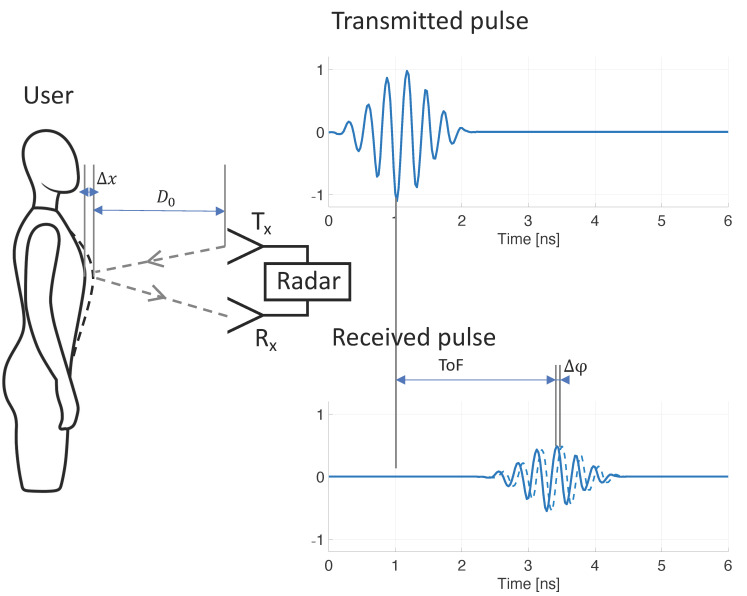
Mechanism of contactless respiratory rate monitoring using ultra-wideband radar. Respiration results in mechanical displacement of the chest and abdominal wall (Δx), which causes variations to the time-of-flight (*ToF*) of the transmitted radar signal. By tracking these variations (Δφ) over time, respiratory rate can be discerned.

**Figure 2 sensors-21-04027-f002:**
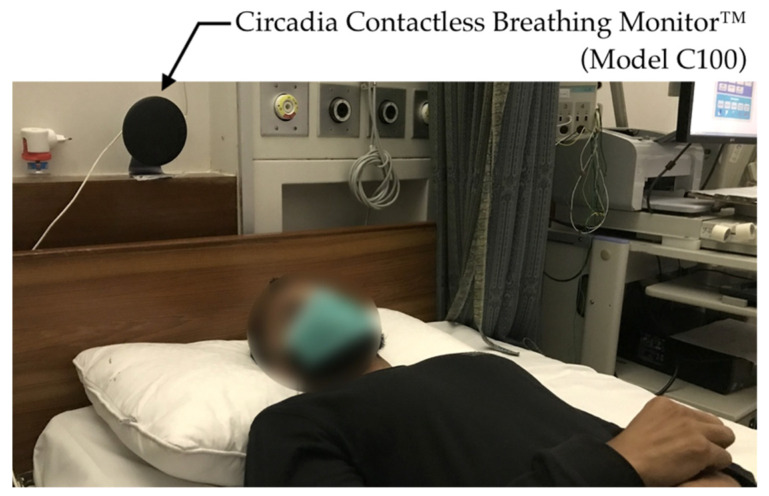
Photo of the Circadia Monitor placement, representative for the experimental setup in Studies 2 and 3.

**Figure 3 sensors-21-04027-f003:**
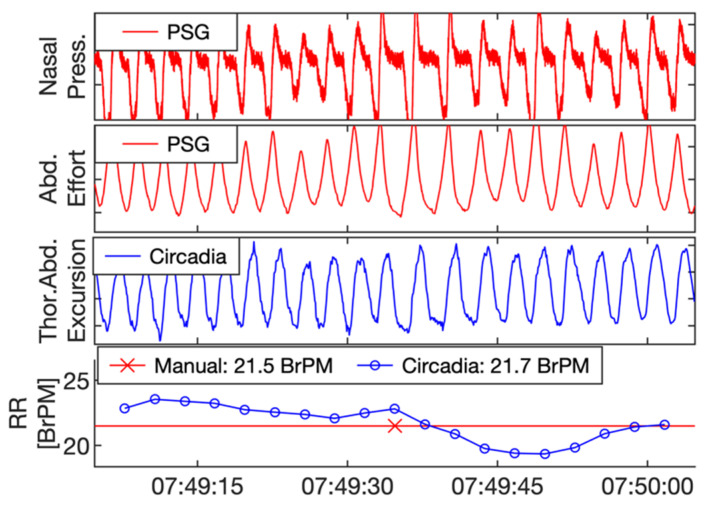
Example of reference polysomnography (PSG) nasal pressure and abdominal effort data (red), and Circadia thoracoabdominal excursion (blue). Each 60 s epoch of PSG data was manually scored for respiratory rate (RR; bottom graph; red). A mean RR value was obtained from continuous Circadia RR data in the corresponding window (bottom graph; blue), allowing for direct comparison of RR. Final RR values of both methods are given in the legend of the bottom graph.

**Figure 4 sensors-21-04027-f004:**
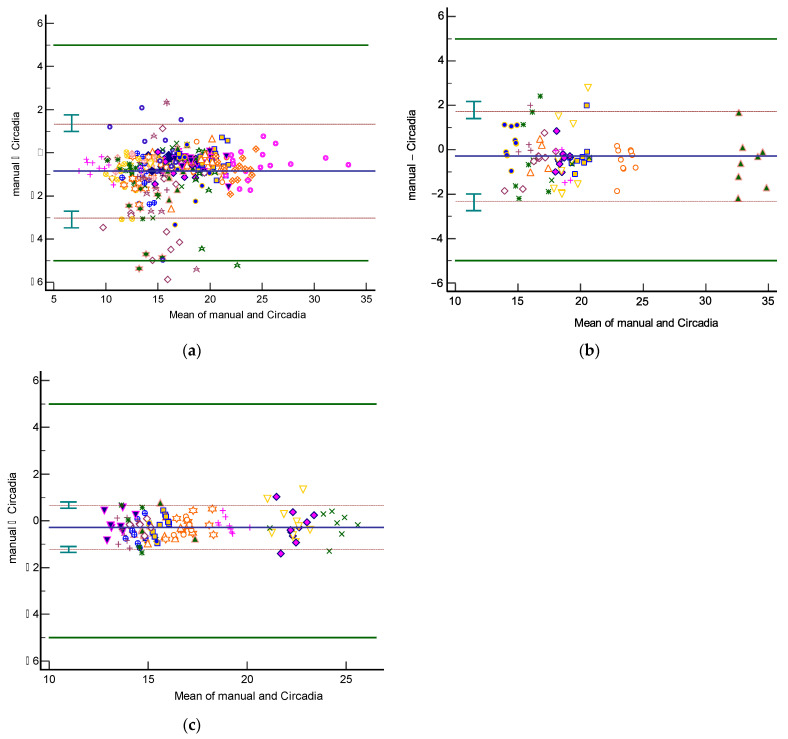
Bland–Altman plots of respiratory rate agreement between Circadia and reference for Study 1 (**a**), Study 2 (**b**), and Study 3 (**c**). The bias (solid blue line) and 95% limits of agreement (LOA; brown dashed lines) are indicated, as well as confidence intervals around the LOA (green error bars). Data from each subject are indicated using a unique marker. The target agreement of ±5 breaths per minute is denoted using solid green lines.

**Figure 5 sensors-21-04027-f005:**
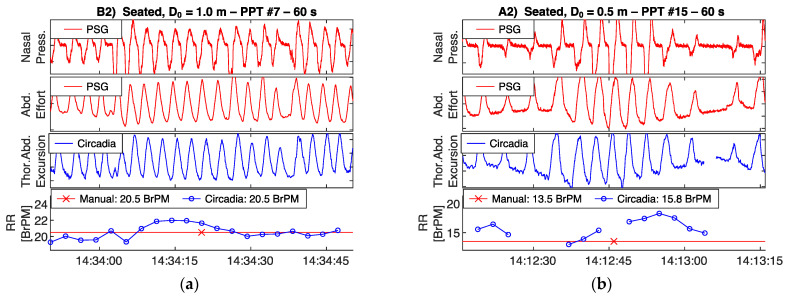
Circadia respiratory thoracoabdominal excursion (blue), and reference nasal pressure and abdominal effort (red) for two example epochs from Study 1. Circadia continuous respiratory rate (RR; blue) and manually scored RR based on reference data (red) are shown in the bottom graph, with final RR values of both methods in the legend of the bottom graph.

**Figure 6 sensors-21-04027-f006:**
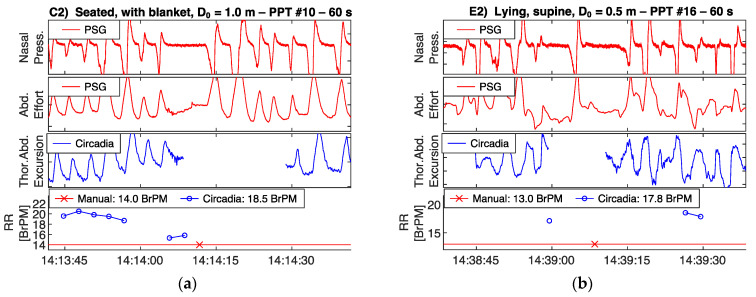
Circadia respiratory thoracoabdominal excursion (blue), and reference nasal pressure and abdominal effort (red) for two example epochs from Study 1. Circadia continuous respiratory rate (RR; blue) and manually scored RR based on reference data (red) are shown in the bottom graph, with final RR values of both methods in the legend of the bottom graph.

**Table 1 sensors-21-04027-t001:** Overview of subject populations and experimental conditions in Study 1, 2, and 3.

	Study 1	Study 2	Study 3
Sample	Nonprobability sample of volunteers, wide range of age, BMI, and sex	Patients in an overnight OSA diagnostic assessment, suspicion of OSA, various health conditions *	Healthy volunteers in an overnight sleep–memory study
*n* (of which women)	26 (10)	12 (4)	12 (11)
Age	23–73 (uniform)	Mean 55.0 (±10.6 SD)	Mean 23.7 (±3.2 SD)
BMI	19.6–60.7 (uniform)	Mean 33.3 (±7.8 SD)	Not recorded
Reference method	Manually scored ventilatory effort (nasal pressure, thoracic and abdominal RIP belts)	Manually scored ETCO_2_ capnography	Manually scored abdominal effort belt
Recording conditions	Seated (1 min, *D*_0_: 0.5 m) Seated (1 min, *D*_0_: 1.0 m) Seated, with blanket (1 min, *D*_0_: 1.0 m) Seated (1 min, *D*_0_: 1.5 m) Lying Supine (1 min, *D*_0_: 0.5 m) Lying Supine, with blanket (1 min, *D*_0_: 0.5 m) Lying Side, facing device (1 min, *D*_0_: 0.5 m) Lying Side, back to device (1 min, *D*_0_: 0.5 m) Lying Supine (9 min, *D*_0_: 0.5 m)	10 min recording, while asleep, with blanket. *D*_0_: 0.5–1.0 m	10 min recording, while asleep, with blanket. *D*_0_: 0.5–1.0 m

OSA: obstructive sleep apnea, *n*: number of subjects, BMI: body mass index, RIP: respiratory inductance plethysmography, ETCO_2_: end-tidal carbon dioxide, *D*_0_: device-to-subject distance * Health conditions included obstructive sleep apnea, hypertension, and diabetes mellitus.

**Table 2 sensors-21-04027-t002:** Overview of results for Studies 1, 2, and 3. Mean metrics are given across subjects, with standard deviation.

	Study 1 (*n* = 26)	Study 2 (*n* = 12)	Study 3 (*n* = 12)
Total Number of Epochs	468	120	150
Number of Technician Excluded Epochs	69	12	6
Mean Circadia Success Rate (±SD)	98.5% (±3.0 pp)	91.6% (±12.7 pp)	100.0% (±0.0 pp)
Mean MAE (±SD)	0.95 (±0.65) BrPM	0.81 (±0.35) BrPM	0.44 (±0.13) BrPM
Mean Accuracy Rate (±SD): Error <2 BrPM	89.7% (±16.6 pp)	94.1% (±9.8 pp)	100.0% (±0.0 pp)
Agreement: Bias	−0.85 BrPM	−0.29 BrPM	−0.27 BrPM
95% LOA Lower (95% CI)	−3.03 (−3.5 to −2.7) BrPM	−2.31 (−2.7 to −2.0) BrPM	−1.21 (−1.4 to −1.1) BrPM
95% LOA Upper (95% CI)	1.32 (1.0 to 1.8) BrPM	1.73 (1.4 to 2.2) BrPM	0.66 (0.5 to 0.8) BrPM

*n*: number of subjects, pp: percentage point, MAE: mean absolute error, BrPM: breaths per minute, LOA: limits of agreement.

**Table 3 sensors-21-04027-t003:** Respiratory rate accuracy results for individual conditions in Study 1.

Condition		Num Epochs	Num Excluded Epochs	Circadia Success Rate (%)	MAE (BrPM)	Accuracy Rate (%)	Bias (BrPM)	LOA (BrPM)
A	Seated, *D*_0_: 0.5 m	52	21	100.0	0.85	90.3	−0.67	−2.25 to 0.91
B	Seated, *D*_0_: 1.0 m	52	6	100.0	0.89	93.5	−0.89	−2.73 to 0.95
C	Seated, *D*_0_: 1.0 m, blanket	52	6	100.0	0.93	91.3	−0.89	−3.00 to 1.22
D	Seated, *D*_0_: 1.5 m	52	5	100.0	0.75	95.7	−0.72	−2.76 to 1.31
E	Lying supine, *D*_0_: 0.5 m	52	5	97.9	1.04	84.8	−0.85	−3.74 to 2.04
F	Lying supine, *D*_0_: 0.5 m, blanket	52	8	90.9	1.04	85.0	−0.99	−3.37 to 1.39
G	Lying side, facing device, *D*_0_: 0.5 m	52	6	100.0	0.63	97.8	−0.62	−1.52 to 0.29
H	Lying side, facing away, *D*_0_: 0.5 m	52	6	100.0	0.94	91.3	−0.99	−3.66 to 1.68
I	Lying supine, *D*_0_: 0.5 m, 9 min	234	30	95.6	0.98	88.2	−0.89	−3.22 to 1.45

MAE: mean absolute error, BrPM: breaths per minute, LOA: limits of agreement, *D*_0_: device-to-subject distance.

## Data Availability

The data presented in this study are available on request from the corresponding author. The data are not publicly available due to intellectual property concerns and the use of proprietary data formats. Some of the data may be of a commercially sensitive nature and a non-disclosure agreement may be required for data access.

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
