# Peer review of "Validation of a New Contactless and Continuous Respiratory Rate Monitoring Device Based on Ultra-Wideband Radar Technology"

_sensors, 2021, doi:10.3390/s21124027_

Round 1
Reviewer 1 Report
Although the general topic is interesting to the community, the novelty and relevance of the work is dubious. It is not focused on a new device or on a new application or technique but only on the performance validation of the Circadia Contactless Breathing Monitor. It seems a commercial device thus I cannot understand what is the relevance for the scientific community.
According to the absence of technical information evidenced in my previous observation, the introduction lacks a proper comparison between the currently employed techniques for the radar-based breathing detection.
A short comparison and citation of the recent alternative solutions for the radar-based breathing detection should be reported and critically described, by highlighting the pros and cons of the different solutions. Some famous examples are Doppler and FMCW vital sign detection but in the literature it is also possible to find examples of biomedical MIMO radars for the vital sign detection.
Also, the general principles of the radar vital sign detection should be disclosed in a theoretical section. Here I reported some recent high-level references which might be useful to perform such analysis and to present the last research findings in this field.
doi: 10.1109/TMTT.2021.3053972.
doi: 10.1109/TMTT.2019.2931834.
doi: 10.1109/TMTT.2021.3049514.
doi: 10.1109/JSEN.2020.3024961.
Minor comment:
From Fig.1 it seems that Δ? is the period of the signal, instead I understand that it is the phase difference between the transmitted and received signals.
Reviewer 2 Report
The authors presented a contactless respiratory rate monitor (Circadia C100). They compared the measurements obtained with manually scored techniques to validate it. Three studies were carried on samples of healthy and diseased people. Authors showed a good agreement and advised the use of the monitor in clinical practice.
Overall, the paper is scientific sound. Clinical relevance and background are well reported and documented. Results are well presented and discussed.
My decision is accepted after minor revision. Here are my amendments to the paper:
- Move the description of the technique (lines 69-76) from the Introduction to the Material and methods section
- Is there a minimum and/or maximum distance for the monitor for study 2 and study 3?
- Is there a preprocessing step for the signals?
- accuracy rate is not well reported for study 2 and study 3
Reviewer 3 Report
The present work aims to validate the performance of the The Circadia Contactless Breathing Monitor™ (model C100): a commercially available device for respiratory rate monitoring via UWB signals. Three studies have been performed for the validation involving 500 subjects and 809 minutes of data recorded.
The presented UWB device for RR analysis is very interesting. However, the validation study should be expanded. The organization of the paper can be improved and the introduction should include more related works. Below, some comments the author should address:
- A comprehensive review of the related works is missing;
- Sections 2.1 and 2.2 could be merged and table1 should summarize both sections;
- The data processing section could be merged into the statistical analysis section;
- Section 2.2.1 and Table1: it is not clear the meaning of “distance”; Distance from what?
- Testing procedures: it is not clear the description of the reference devices which are used to compare the performance of the proposed device. For example, when describing “Thoracic effort, abdominal effort, nasal pressure, and nasal flow were obtained using the 144 Natus Embletta MPR PG (Pleasanton, USA) polygraphy device.”, it is not clear how the considered parameters can be connected to the respiratory rate. Please clarify the connection of the respiratory rate to the measurements of Thoracic effort, abdominal effort, nasal pressure?
- Fig.2: please add axis labels;
- Fig.3 could be eliminated. The relevant information can be included in the text;
- Fig.4c presents overlapping writings;
- Most of the figures present a long caption including a part of the discussion. Please move the discussion to the discussion section;
- What is the impact of the distance between the device and the subject on the measurement accuracy?
- What about the sensitivity of the device with respect to the subject's mobility?
- A picture illustrating the system setup could be appreciated;
- Please expand the conclusion. What does “reference data was high and within the target agreement” mean?
- Please expand the discussion about the results related to different body positions shown in Table 2.
Round 2
Reviewer 1 Report
The authors addressed all my concerns.
Reviewer 3 Report
The author has properly addressed the reviewer's comments.